# Cross-client Label Propagation for Transductive and Semi-Supervised Federated Learning

**Jonathan Scott**                                      *jonathan.scott@ist.ac.at*
*ISTA (Institute of Science and Technology Austria), Klosterneuburg, Austria*

**Michelle Yeo**                                         *michelle.yeo@ist.ac.at*
*ISTA (Institute of Science and Technology Austria), Klosterneuburg, Austria*

**Christoph H. Lampert**                                      *chl@ist.ac.at*
*ISTA (Institute of Science and Technology Austria), Klosterneuburg, Austria*

**Reviewed on OpenReview:** *https://openreview.net/forum?id=gYO4GX8R5k*

## Abstract

We present *Cross-Client Label Propagation (XCLP)*, a new method for transductive and semi-supervised federated learning. XCLP estimates a data graph jointly from the data of multiple clients and computes labels for the unlabeled data by propagating label information across the graph. To avoid clients having to share their data with anyone, XCLP employs two cryptographically secure protocols: *secure Hamming distance computation* and *secure summation*. We demonstrate two distinct applications of XCLP within federated learning. In the first, we use it in a one-shot way to predict labels for unseen test points. In the second, we use it to repeatedly pseudo-label unlabeled training data in a federated semi-supervised setting. Experiments on both real federated and standard benchmark datasets show that in both applications XCLP achieves higher classification accuracy than alternative approaches.

## 1 Introduction

Federated Learning (FL) (McMahan et al., 2017) is a machine learning paradigm in which multiple clients, each owning their own data, cooperate to jointly solve a learning task. The process is typically coordinated by a central server. The defining restriction of FL is that client data must remain on device and cannot be shared with either the server or other clients. In practice this is usually not due to the server being viewed as a hostile party but rather to comply with external privacy and legal constraints that require client data to remain stored on-device. To date, the vast majority of research within FL has been focused on the supervised setting, in which client data is fully labeled and the goal is to train a predictive model. In this setting a well-defined template has emerged: first proposed as *federated averaging* (McMahan et al., 2017), this consists of alternating between local model training at the clients and model aggregation at the server.

However, in many real-world settings fully labeled data may not be available. For instance, in *cross-device* FL, smartphone users are not likely to be interested in annotating more than a handful of the photos on their devices (Song et al., 2022). Similarly, in a *cross-silo* setting the labeling of medical imaging data may be both costly and time consuming (Dehaene et al., 2020). As such, in recent years there has been growing interest in developing algorithms that can learn from partly labeled or fully unlabeled data in a federated setting (Kairouz et al., 2021). For such algorithms it can be beneficial, or even essential, to go beyond the standard federated framework of model-centric learning and develop techniques that directly leverage client data interactions. Examples include federated clustering (Dennis et al., 2021) and dimensionality reduction (Grammenos et al., 2020), where clients compute statistics based on their data and the server computes with aggregates of these statistics.

**Contribution**   In this work we propose *Cross-Client Label Propagation (XCLP)* which follows a data-centric approach. XCLP allows multiple clients, each with labeled and unlabeled data, to cooperatively compute labels for their unlabeled data. This is done using a transductive approach, where the goal is label inference restricted to some predefined set of unlabeled examples. In particular this approach does not require a model to be trained in order to infer labels. Specifically, XCLP takes a graph-based approach to directly assign labels to the unlabeled data. It builds a joint data graph of a group of clients and propagates the clients' label information along the edges. Naively, this approach would require the clients to centrally share their data. That, however, would violate the constraints of federated learning.

XCLP allows for multiple clients to jointly infer labels from a cross-client data graph *without* them having to share their data. We refer to this privacy guarantee, in which a client's data never leaves their own device, as data confidentiality. To achieve this XCLP exploits the modular and distributed nature of the problem. It uses locality-sensitive hashing and secure Hamming distance computation to efficiently estimate the cross-client data graph. It then distributes the label propagation computation across clients and aggregates the results using a customized variant of *secure summation*. The key benefits of this approach are:

- XCLP enables the data of multiple clients to be leveraged when estimating the graph and propagating labels. This is beneficial as the prediction quality of label propagation increases substantially when more data is used.

- XCLP achieves data confidentiality, since it does not require clients to share their data with anyone else. Instead clients only have to share approximate data point similarities with the server.

- XCLP is communication efficient as it does not require training a model over multiple communication rounds but requires only a single round of communication.

As a technique to transfer label information from labeled to unlabeled points, XCLP is versatile enough to be used in a variety of contexts. We illustrate this by providing two applications within federated learning. In the first we employ XCLP purely for making predictions at inference time. We demonstrate empirically on a real-world, highly heterogeneous, federated medical image dataset that XCLP is able to assign high quality labels to unlabeled data. When all clients are partly labeled we observe XCLP to outperform purely local label propagation, which illustrates the benefits of leveraging more data. XCLP also obtains strong results when using fully labeled clients to infer labels on different, fully unlabeled clients, even when these clients have very different data distributions. In both these scenarios we find that running XCLP on features obtained from models trained using *FederatedAveraging* gives significantly better accuracy than purely using the model predictions.

In the second application we tackle the problem of federated semi-supervised learning. In this scenario clients possess only partly labeled data and the goal is to train a classifier by leveraging both labeled and unlabeled client data. In this setting we employ XCLP in the training process by integrating it into a standard federated learning pipeline. Specifically, during each round of *FederatedAveraging* we use XCLP to assign pseudo-labels and weights to the unlabeled data of the current batch of clients. These are then used to train with a weighted supervised loss function. Our experiments show that this pseudo-labelling approach outperforms all existing methods for federated semi-supervised learning, as well as a range of natural baselines in the standard federated CIFAR-10 benchmark. Going beyond prior work, we also evaluate on more challenging datasets, namely CIFAR-100 and Mini-ImageNet, where we also observe substantial improvements in accuracy.

## 2   Related Work

**Federated Learning**   Federated learning (FL) (McMahan et al., 2017) was originally proposed for learning on private fully labeled data split across multiple clients. For a survey on developments in the field see (Kairouz et al., 2021). A number of recent works propose federated learning in the absence of fully labeled data. Methods for cluster analysis and dimensionality reduction have been proposed (Dennis et al., 2021; Grammenos et al., 2020), in which the server acts on aggregates of the client data, as opposed to client models. Other works have focused on a model based approach to federated semi-supervised learning (SSL). Jeong

---

**Algorithm 1:** `Cross-ClientLabelPropagation`

---

**Input:** set of participating clients $P$, client data $(V^{(j)}, Y^{(j)})_{j \in P}$      // *data stored on-device at clients*

1-XS: Setup: clients exchange private and public keys, agree on random seed $s$

2-CS: **for** *client $j \in P$ in parallel* **do**

3-CS: $\quad \Pi \in \mathbb{R}^{L \times d} \quad$ with $\quad \Pi_{ij} \overset{i.i.d.}{\sim} \mathcal{N}_{\text{seed}=s}(0,1)$      // *same $\Pi$ for each client*

4-CS: $\quad B^{(j)} \leftarrow \text{sign}(\Pi\,(V^{(j)})^{\top})$      // *LSH projection*

5-XS: $H \leftarrow \texttt{SecureHamming}((B^{(j)})_{j \in P})$      // *server gets Hamming matrix*

6-SS: $A \leftarrow \cos(\frac{\pi}{L} H)$      // *estimate cosine similarity matrix*

7-SS: $B \leftarrow \text{sparsify}(A)$      // *keep $k$ largest entries per row, set others to 0*

8-SS: $W = B + B^{\top}$      // *symmetrize*

9-SS: $\mathcal{W} \leftarrow D^{-\frac{1}{2}} W D^{-\frac{1}{2}}$ for $D = \text{diag}(d_1, \ldots, d_n)$ with $d_i = \sum_j W_{ij}$      // *normalize*

10-SS: $S \leftarrow (\text{Id}_{n \times n} - \alpha \mathcal{W})^{-1}$      // *influence matrix*

11-CS: **for** *client $j \in P$ in parallel* **do**

12-CS: $\quad S_L^{(j)} \leftarrow \text{labeled-cols}_j(S)$      // *client gets columns corresponding to labeled data*

13-CS: $\quad \bar{Z}^{(j)} \leftarrow S_L^{(j)} Y_L^{(j)}$      // *compute local contribution to overall label propagation*

14-XS: $\quad Z^{(j)} \leftarrow \texttt{SecureRowSums}\big((\bar{Z}^{(k)})_{k \in P}\big)_j$      // *client gets its part of aggregated contributions*

15-CS: $\quad \hat{y}^{(j)} \leftarrow \big(\arg\max_{c=1,\ldots,C} Z_{i,c}^{(j)}\big)_{i=1,\ldots,n^{(j)}}$      // *predict labels*

16-CS: $\quad \omega^{(j)} \leftarrow \Big(1 - \texttt{entropy}\big(\frac{(Z_{i,c}^{(j)})_{c=1,\ldots,C}}{\sum_{c=1}^{C} Z_{i,c}^{(j)}}\big)/\log C\Big)_{i=1,\ldots,n^{(j)}}$      // *predicted label confidences*

**Output:** predicted labels and confidences $(\hat{y}^{(j)}, \omega^{(j)})_{j \in P}$      // *available only to respective clients*

---

et al. (2021) propose inter-client consistency and parameter decomposition to separately learn from labeled and unlabeled data. Long et al. (2020) apply consistency locally through client based teacher models. Liang et al. (2022) combine local semi-supervised training with an enhanced aggregation scheme which re-weights client models based on their distance from the mean model. Zhang et al. (2021b) and Diao et al. (2022) focus on a setting in which the server has access to labeled data. In this setting Zhang et al. (2021b) combine local consistency with grouping of client updates to reduce gradient diversity while Diao et al. (2022) combine consistency, through strong data augmentation, with pseudo-labeling unlabeled client data. Our approach to federated SSL is to iteratively apply XCLP to pseudo-label unlabeled client data. This approach differs from prior work by making use of data interactions between multiple clients to propagate label information over a cross client data graph. Related to this idea is the notion of federated learning on graphs (Zhang et al., 2021a; Xie et al., 2021; Wang et al., 2020). However, these works are primarily interested in learning from data that is already a graph. In contrast XCLP estimates a graph based on similarities between data points, in order to spread label information over the edges.

**Label Propagation**    Label Propagation (Zhu & Ghahramani, 2002; Zhou et al., 2004) was originally proposed as a tool for transductive learning with partly labeled data. Over time it has proven to be a versatile tool for a wide range of problems. Several works have applied LP to the problem of domain adaptation. Liu et al. (2019) apply LP over a learned graph for few shot learning. Cai et al. (2021) develop a framework for domain adaptation by combining LP with a teacher trained on the source. Khamis & Lampert (2014) use LP as a prediction-time regularizer for collective classification tasks. In the context of deep semi-supervised learning Iscen et al. (2019) make use of LP as a means of obtaining pseudo-labels for unlabeled data which are then used in supervised training. Several works apply LP to problems with graphical data. Huang et al. (2021) observe that combining linear or shallow models with LP can lead to performance that is on par with or better than complex and computationally expensive GNNs. Wang & Leskovec (2022) apply LP as a regularizer for graph convolutional neural networks when learning edge weights and quantify the theoretical connection between them in terms of smoothing.

## 3 Method

In this section we begin by introducing the problem setting. In Section 3.1 we present our method XCLP, in Section 3.2 we describe the cryptographic protocols we make use of and in Section 3.3 we provide an analysis of XCLP.

Let $P$ be a set of client devices for which labels should be propagated. Note that we place no restrictions on $P$. It could be all clients in a federated learning scenario, a randomly chosen subset, or a strategically chosen subset, e.g. based on client similarity, diversity or data set sizes.

Each client $j \in P$ possesses a set of $n^{(j)}$ $d$-dimensional data vectors, of which $l^{(j)}$ are labeled, i.e. $V^{(j)} = (v_1^{(j)}, \ldots v_{l^{(j)}}^{(j)}, v_{l^{(j)}+1}^{(j)}, \ldots v_{n^{(j)}}^{(j)})^\top \in \mathbb{R}^{n^{(j)} \times d}$, with partial labels $\{y_1^{(j)}, \ldots, y_{l^{(j)}}^{(j)}\}$ from $C$ classes, which we encode in zero-or-one-hot matrix form: $Y^{(j)} \in \{0,1\}^{n^{(j)} \times C}$ with $Y_{ic}^{(j)} = \mathbb{1}\{y_i^{(j)} = c\}$ for $1 \le i \le l^{(j)}$ and $Y_{ic}^{(j)} = 0$ otherwise. Note that this setup includes the possibility for a client to have only labeled data, $n^{(j)} = l^{(j)}$ or only unlabeled data, $l^{(j)} = 0$. We denote the total amount of data by $n := \sum_{j \in P} n^{(j)}$. Our goal is to assign labels to the unlabeled data points, i.e., *transductive learning* (Vapnik, 1982). The process is coordinated by a central server, which we assume to be *non-hostile*. That means, we trust the server to operate on non-revealing aggregate data and to return correct results of computations.[1] At the same time, we treat clients and server as *curious*, i.e., we want to prevent that at any point in the process any client's data is revealed to the server, or to any other client.

Our main contribution in this work is, *Cross-Client Label Propagation (XCLP)*, an algorithm for assigning labels to the unlabeled data points. XCLP works by propagating label information across a neighborhood graph that is built jointly from the data of all participating clients without revealing their data. Before explaining the individual steps in detail, we provide a high level overview of the method.

XCLP consists of three phases: 1) the clients jointly compute similarity values between all of their data points and transfer them to the server, 2) the server uses these similarities to construct a neighborhood graph and infers an influence matrix from this, which it distributes back to the clients, 3) the clients locally compute how their data influences others, aggregate this information, and infer labels for their data.

The key challenge is how to do these steps without the clients having to share their data and labels with each other or with the server. XCLP manages this by formulating the problem in a way that allows us to use only light-weight cryptographic protocols for the steps of computing similarities and aggregating label information.

### 3.1 Cross-Client Label Propagation (XCLP)

Algorithm 1 shows pseudocode for XCLP. To reflect the distributed nature of XCLP we mark the execution type of each step: *client steps (CS)* are steps that clients do locally using only their own data, *server steps (SS)* are steps that the server executes on aggregated data, *cross steps (XS)* are steps that require cross-client or client-server interactions.

As a setup step (line 1) the clients use a secure key exchange procedure to agree on a shared random seed that remains unknown to the server. This is a common step in FL when cryptographic methods, such as *secure model aggregation*, are employed, see Bonawitz et al. (2017).

**Phase 1.** The clients use the agreed-on random seed to generate a common matrix $\Pi \in \mathbb{R}^{L \times d}$ with unit Gaussian random entries (line 3). Each client, $j$, then multiplies its data matrix $V^{(j)}$ by $\Pi$ and takes the component-wise sign of the result, thereby obtaining a matrix of $n^{(j)}$ $L$-dimensional binary vectors, $B^{(j)}$ (line 4). In combination, both steps constitute a local *locality-sensitive hashing (LSH)* (Indyk & Motwani, 1998) step for each client. A crucial property of this encoding is that the (cosine) similarity between any two data vectors, $v, v'$, can be recovered from their binary encodings $b, b'$: $\text{sim}(v, v') := \frac{\langle v, v' \rangle}{\|v\| \|v'\|} \approx \cos(\pi h(b, b')/L)$, where $h(b, b') = \sum_{l=1}^{L} b_l \oplus b_l'$ is the Hamming distance (number of bits that differ) between binary vectors

---

[1]In particular we exclude *malicious* servers in the cryptographic sense that would, e.g., be allowed to employ attacks such as model poisoning or generating fake clients in order to break the protocol.

and $\oplus$ is the XOR-operation. Since all clients use identical random projections, this identity holds even for data points located on different clients. See Appendix C.2 for details on LSH.

In line 5 the necessary Hamming distance computations take place using a cryptographic subroutine that we detail in Section 3.2. Note that cryptographic protocols operate most efficiently on binary vectors, and Hamming distance is particularly simple to compute. In fact, this is the reason why we transform the data using LSH in the first place. Ultimately, from this step the server obtains the matrix of all pairwise Hamming distances, $H \in \mathbb{Z}^{n \times n}$, but no other information about the data.

**Phase 2.** Having obtained $H$ the server executes a number of steps by itself. First, it converts $H$ to a (cosine) similarity matrix $A \in \mathbb{R}^{n \times n}$ (line 6). It sparsifies each row of $A$ by keeping the $k$ largest values and setting the others to 0 (line 7). From the resulting matrix, $B$, it constructs a weighted adjacency matrix of the data graph, $\mathcal{W}$ by symmetrization (line 8) and normalization (line 9).

If not for the aspect of data confidentiality, we could now achieve our goal of propagating label information along the graph edges from labeled to unlabeled points in the following way: form the concatenation of all partial label matrices, $Y = (Y^{(j)})_{j \in P} \in \{0, 1\}^{n \times C}$, and compute $Z = SY \in \mathbb{R}^{n \times C}$, where $S = (\mathrm{Id} - \alpha \mathcal{W})^{-1}$ is the *influence matrix*, and $\alpha \in (0, 1)$ is a hyperparameter. See Appendix C.1 for an explanation how this step corresponds to the propagation of labels over the graph.

XCLP is able to perform the computation of the *unnormalized class scores*, $Z$, without having to form $Y$, thereby preserving the confidentiality of the labels. Instead, it computes only the influence matrix, $S$, centrally on the server (line 10), while the multiplication with the labels will be performed in a distributed way across the clients.

**Phase 3.** Observe that the computation of $Z$ can also be written as $Z = \sum_{j \in P} S^{(j)} Y^{(j)}$, where $S^{(j)} \in \mathbb{R}^{n \times n^{(j)}}$ is the sub-matrix of $S$ consisting of only the columns that correspond to the data of client $j$. We can refine this further, note that all rows of $Y^{(j)}$ that correspond to the unlabeled data of client $j$ are identically 0 by construction and hence do not contribute to the multiplication. Therefore, writing $Y_L^{(j)} \in \mathbb{R}^{l^{(j)} \times C}$ for the rows of $Y^{(j)}$ that correspond to labeled points and $S_L^{(j)} \in \mathbb{R}^{n \times l^{(j)}}$ for the corresponding columns of $S^{(j)}$, it also holds that $Z = \sum_{j \in P} S_L^{(j)} Y_L^{(j)}$.

Using this observation, Algorithm 1 continues by each client $j$ receiving $S_L^{(j)}$ from the server (line 12). It then locally computes $\bar{Z}^{(j)} = S_L^{(j)} Y_L^{(j)} \in \mathbb{R}^{n \times C}$ (line 13), which reflects the influence of $j$'s labels on all other data points.

By now, the clients have essentially computed $Z$, but the result is additively split between them: $Z = \sum_{j \in P} \bar{Z}^{(j)}$. To compute the sum while preserving data confidentiality, XCLP uses a secure summation routine (line 14), as commonly used in FL for model averaging (Bonawitz et al., 2017). However, to increase its efficiency we tailor it to the task, see Section 3.2 for details. As a result, each client $j$ receives only those rows of $Z$ that correspond to its own data, $Z^{(j)} \in \mathbb{R}^{n^{(j)} \times C}$ (line 13). From these, it computes labels and confidence values for its data (lines 15-16).

### 3.2 Cryptographic Subroutines

In this section we describe the proposed cryptographic protocols for *secure summation* and *secure Hamming distance computation* that allow clients to run XCLP without having to share their data.

**SecureRowSums** We propose a variation of the *secure summation* that is commonly used in FL (Bonawitz et al., 2017). For simplicity we describe the case where all values belong to $\mathbb{Z}_l$ for some $l \in \mathbb{N}$, though extensions to fixed-point or floating-point arithmetic also exist (Catrina & Saxena, 2010; Aliasgari et al., 2013).

Given a set of clients $P$, each with some matrix $Z^{(j)} \in \mathbb{Z}_l^{n \times c}$, ordinary **SecureSum** computes $\sum_{j \in P} Z^{(j)}$ at the server in such a way that the server learns only the sum but nothing about the $Z^{(j)}$ matrices. The main idea is as follows: using agreed upon random seeds clients jointly create random matrices $M^{(j)} \in \mathbb{Z}_l^{n \times c}$ with the property that $\sum_{j \in P} M^{(j)} = 0$. Each client $j$ then obfuscates its data by computing $\tilde{Z}^{(j)} := Z^{(j)} + M^{(j)}$

and sends this to the server. From the perspective of the server each $\tilde{Z}^{(j)}$ is indistinguishable from uniformly random noise. However, when all parts are summed, the obfuscations cancel out and what remains is the desired answer: $\sum_{j \in P} \tilde{Z}^{(j)} = \sum_{j \in P} Z^{(j)}$. For technical details see Bonawitz et al. (2017).

For XCLP, we propose a modification of the above construction. Suppose we have a partition of the rows, $(R_j)_{j \in P}$, where each $R_j \subset [n]$. Each client $j$ knows its own $R_j$ and the server knows all $R_j$. `SecureRowSums`'s task is to compute $Z := \sum_{j \in P} Z^{(j)}$ in a distributed form in which each client $j \in P$ learns only the rows of $Z$ indexed by $R_j$, denoted $Z[R_j]$, and the server learns nothing.

For this, let $M^{(j)}$ and $\tilde{Z}^{(j)}$ be defined as above. In addition, let $\hat{Z}^{(j)}$ be equal to the obfuscated $\tilde{Z}^{(j)}$, except that the rows indexed by $R_j$ are completely set to 0. Each client now instead sends $\hat{Z}^{(j)}$ to the server, which computes $\hat{Z} := \sum_{j \in P} \hat{Z}^{(j)}$. The server then redistributes $\hat{Z}$ among the clients, i.e. each client $j$ receives $\hat{Z}[R_j]$. Note that $\hat{Z}[R_j] = \sum_{k \in P \setminus \{j\}} \tilde{Z}^{(k)}[R_j] = Z[R_j] - \tilde{Z}^{(j)}[R_j]$. Consequently, each client obtains the part of $Z$ corresponding to its own data by computing $Z[R_j] = \hat{Z}[R_j] + \tilde{Z}^{(j)}[R_j]$. By construction the shared quantities leak nothing to the server. Specifically $\hat{Z}^{(j)}$ is random noise with rows $R_j$ set to 0 and $\hat{Z}$ is random noise since each block $\hat{Z}[R_j]$ remains obfuscated due to client $j$ not sending $M^{(j)}[R_j]$.

**`SecureHamming`** Several cryptographic protocols for computing the Hamming distances between binary vectors exist. Here we propose two variants that are tailored to the setting of XCLP: one is based on the SHADE protocol (Bringer et al., 2013), which is easy to implement as it only requires only an *oblivious transfer (OT)* (Naor & Pinkas, 2001) routine as cryptographic primitive. The other is based on *partially homomorphic encryption (PHE)* and comes with lower communication cost.

*OT-based secure Hamming distance:* Let $b = (b_1, \ldots, b_L)$ and $b' = (b'_1, \ldots, b'_L)$ be the bit vectors, for which the Hamming distance should be computed, where $b$ is stored on a client $j$ and $b'$ on a client $k$.

1) client $j$ creates $L$ random numbers $r_1, \ldots, r_L$ uniformly in the range $[0, L-1]$. For each $l = 1, \ldots, L$, it then offers two values to be transferred to client $k$: $z_l^0 = r_l + b_l$ or $z_l^1 = r_l + \bar{b}_l$, for $\bar{b}_l = 1 - b_l$ (here and in the following all calculations are performed in $\mathbb{Z}_L$, i.e. in the integers *modulo L*).

2) Client $k$ initiates an OT operation with input $b'_l$ and result $t_l$. That means, if $b'_l = 0$ it will receive $t_l = z_l^0$ and if $b'_l = 1$ it will receive $t_l = z_l^1$, but client $i$ will obtain no information which of the two cases occurred. Note that in both cases, it holds that $t_l = r_l + b_l \oplus b'_l$. However, client $k$ gains no information about the value of $b_l$ from this, because of the uniformly random shift $r_l$ that is unknown to it.

3) Clients $j$ and $k$ now compute $R = \sum_{l=1}^L r_l$ and $T = \sum_{l=1}^L t_l$, respectively, and send these values to the server.

4) From $R$ and $T$, the server can infer the Hamming distance between $b$ and $b'$ as $T - R = \sum_{l=1}^L b_l \oplus b'_l = h(b, b')$.

Performing these steps for all pairs of data points, the server obtains the Hamming matrix, $H \in \mathbb{N}^{n \times n}$, but no other information about the data. The clients obtain no information about each others' data at all during the computation.

*PHE-based secure Hamming distance:* Homomorphic encryption is a framework that allows computing functions on encrypted arguments (Acar et al., 2018). The outcome is an encryption of the value that would have been the result of evaluating the target function on the plaintext arguments, but the computational devices gain no information about the actual plaintext. *Fully homomorphic encryption(FHE)*, which allows arbitrary functions to be computed this way, is not efficient enough for practical usage so far (Jiang & Ju, 2022). However, efficient implementation exist for the setting where cyphertexts only have to be added or subtracted, for example *Paillier*'s (Paillier, 1999). We exploit this paradigm of *partially homomorphic encryption (PHE)* to compute the Hamming distances of a binary vector from clients $j$ with all binary vectors from a client $k$ in the following way:

1) Client $j$ encrypts its own data vector $b^{(j)} \in \{0, 1\}^L$, using its own public key and transfers the resulting vector $\boxed{y} := \mathrm{enc}(b^{(j)})$ (boxes indicate encrypted quantities) to client $k$. Because the data is encrypted, client $k$ can extract no information about client $j$'s data from this.

2) For any of its own data vectors $b^{(k)} \in \{0,1\}^L$, client $k$ creates a uniformly random value $r \in [0, L-1]$. It then computes the following function in a homomorphic way with encrypted input $y = \boxed{y}$ and plaintext input $x = b^{(k)}$:

$$f(y; x) = r + \sum_{l:x_l=1} y_l - \sum_{l:x_l=0} y_l$$

Because of the identity $h(x,y) = \sum_l [x_l y_l + \bar{x}_l \bar{y}_l] = \sum_{l:x_l=1} y_l + \sum_{l:x_l=0} \bar{y}_l = \sum_{l:x_l=1} y_l - \sum_{l:x_l=0} y_l + n - \sum_l x_l$, the result, $\boxed{T} = f(\boxed{y}; x)$, is the value $T = h(b^{(j)}, b^{(k)}) + r - n + \sum_l b_l$ in encrypted form.

4) Client $j$ send the value $R = r - n + \sum_l b_l$ to the server, and the value $\boxed{T}$ to client $j$. Client $j$ decrypts $\boxed{T}$ using its own secret key and sends the resulting value $T$ to the server. It gains no information about client $k$'s data or the Hamming distance, because the added randomness gives $T$ a uniformly random distribution.

5) The server recovers $H(b^{(j)}, b^{(k)}) = T - R$ without gaining any other information about the clients' data.

Note that for computing single Hamming distances, this protocol has no advantage over the OT-based variant. However, for computing all pairwise Hamming distances between the data of two clients, the PHE-based protocol saves a factor $n^{(k)}$ in communication cost, because $\boxed{y}$ has to be transferred only once and can then be used repeatedly by client $k$ to compute the distances to all of its vectors.

### 3.3 Analysis

In this section, we analyze the *efficacy, privacy, efficiency* and *robustness* of Algorithm 1.

**Efficacy**  Algorithm 1 performs label propagation along the data graph, as the classical LP algorithm (Zhu, 2005) does when data is centralized. The similarity measure used is cosine similarity estimated via the Hamming distance of the LSH binary vectors. How close this is to the actual cosine similarity is determined by $L$, the LSH vector length. In practice, we observe no difference in behavior between them already for reasonably small values, e.g. $L = 4096$.

**Privacy**  The main insight is that Algorithm 1 adheres to the federated learning principle that clients do not have to share their data or labels with any other party. This is ensured by the fact that all *cross-steps* are computed using cryptographically secure methods. There are two potential places where clients share some information relating to their data. The first is the matrix of Hamming distances $H$ that is sent to the server, and from which the server can approximately recover the matrix of cosine similarities $W$. While certainly influenced by the client data, we consider $W$ (and therefore $H$) a rather benign object for a non-hostile server to have access to because cosine similarity depends only on angles, hence any rescaling and rotation of the client input vectors would result in the same $W$ matrix. A second source of information sharing is during phase 3 where each client receives the columns of $S$ that correspond to their data. Such columns reflect how their labeled data can influence all other data points according to the data graph as estimated from the participating clients' vectors. However, we stress that this influence is unnormalized and hence the influence relative to other clients cannot be known.

**Computational Complexity**  The computational cost of XCLP is determined by factors: the cryptographic subroutines and the numeric operation. The contribution of the former depends heavily on the underlying implementation and available hardware support, so we do not discuss it here. The cost of the numeric operations can be derived in explicit form. Assume that the number of clients per batch is $p$. Each client has $n$ data points in total, out of which $m$ are labeled (for simplicity, we assume $n$ and $m$ to be identical across clients here). Let the feature dimensionality be $d$ and the number of classes $C$. Then, to run XCLP with $L$-dimensional bit vectors, each client computes an $n \times L$ binary data matrix, which has complexity $O(dnL)$. Then, the clients jointly compute all $p^2 n^2 / 2$ pairwise Hamming distances, which requires $O(Lp^2 n^2)$ operations and has overall complexity $O(Lpn^2)$ if run in parallel by the clients. The server inverts the matrix at cost $O(p^3 n^3)$. Then, each client multiplies an $pn \times m$ sized part of the resulting matrix with their $m \times C$ label matrix, which costs $O(Cpnm)$ per client, and also has overall complexity $O(Cpnm)$ if run on the clients in parallel. Overall, the most costly step is the matrix inversion, but that is done on the server, which we assume to have much higher compute capabilities. Assuming $L > C$ and $np > d$, the clients costs are dominated by the $O(Lpn^2)$ term.

Table 1: XCLP for prediction (Fed-ISIC2019 dataset): Classification accuracy [in %] with two different preprocessing functions (pretrained and finetuned) and training sets of different size (average and standard deviation across three runs).

| training set size | pretrained | | finetuned | | |
| | per-client LP | XCLP | FedAvg | per-client LP | XCLP |
|---|---|---|---|---|---|
| $n = 954$ | $30.70 \pm 1.25$ | $\mathbf{34.00 \pm 1.12}$ | $43.78 \pm 0.88$ | $45.35 \pm 1.25$ | $\mathbf{47.48 \pm 1.12}$ |
| $n = 1882$ | $31.40 \pm 1.18$ | $\mathbf{34.61 \pm 0.93}$ | $46.83 \pm 0.75$ | $48.10 \pm 1.18$ | $\mathbf{52.34 \pm 0.93}$ |
| $n = 3744$ | $37.41 \pm 0.20$ | $\mathbf{38.49 \pm 0.88}$ | $52.87 \pm 0.14$ | $56.98 \pm 0.20$ | $\mathbf{59.49 \pm 0.88}$ |
| $n = 18597$ | $\mathbf{55.10 \pm 0.51}$ | $53.38 \pm 0.67$ | $63.78 \pm 0.20$ | $73.30 \pm 0.51$ | $\mathbf{74.22 \pm 0.67}$ |

For comparison, to run per-client LP, each client has to compute $n^2/2$ $d$-dimensional inner products, invert the resulting $n \times n$ matrix, and multiply a sub-matrix of size $m \times n$ it with a label matrix of size $m \times C$. The complexity per client is $O(dn^2 + n^3 + Cmn)$, which is also the total complexity, if all clients can operate in parallel. In practice, we expect $d > n > C$, so the dominant term is $O(dn^2)$. The server does not contribute any computation. Consequently, with a typical trade-off of, e.g. $L = 8d$, XCLP requires $8p$ times more numeric operations than local LP. Note, however, that one has good control over the total cost, as $L$ and $p$ (and potentially $d$) are design choices.

On an absolute scale,, all of these values are rather small. For example, with $p = 10$, $d = 512$, and $L = 4096$, the number of numeric operations a client has to perform per datapoint is in the order of $10^7$. In many settings, this can be expected to be less than the the operations needed to compute the datapoints feature representation in the first place, e.g. when using a (even small) neural network for that purpose.

**Communication Efficiency** XCLP incurs communication costs at two steps of Algorithm 1. With the OT-based protocol for computing the $n^{(j)} \times n^{(k)}$ Hamming matrix between two clients $j$ and $k$, client $j$ sends $n^{(j)}n^{(k)}L$ integer values in $\mathbb{Z}_L$ to client $k$. With the enhanced PHE-based protocol, this amount is reduced to sending $n^{(j)}L$ encrypted values from $j$ to $k$ and $n^{(j)}n^{(k)}$ in the opposite direction. Each of the two clients sends $n^{(j)}n^{(k)}$ integer values in $\mathbb{Z}_L$ to the server. To propagate the labels via the distributed matrix multiplication, each client $j$ first receives from the server a matrix of size $n \times l^{(j)}$. It transmits a matrix of size $n \times C$ to the server, and receives a matrix of size $n^{(j)} \times C$ back from it. In particular, XCLP requires only a constant number of communication steps, which is in contrast to other methods that train iteratively.

**Robustness** In *cross-device* FL clients may be unreliable and prone to disconnecting spontaneously. Therefore, it is important that FL algorithms can still execute even in the event of intermediate client dropouts. This is indeed the case for Algorithm 1: a client dropping out before the `SecureHamming` step (line 5), is equivalent to it not having been in $P$ in the first place. Since the Hamming computation is executed pairwise, a client dropping out during this step has no effect on the computation of other clients. The result will be missing entries in the matrix, $H$, which the server can remove, thereby leading to the same outcome as if the client had dropped out earlier. If clients drop out after $H$ has been computed, but before the `SecureRowSums` step (line 14), they will have contributed to the estimate of the data graph, but they will not contribute label information to the propagation step. This has the same effect as if the client only had unlabeled data. If clients drop out within `SecureRowSums`, after the obfuscation matrices have been agreed on but before the server has computed $\hat{Z}$, then the secure summation could not be completed. To recover, the server can simply restart the `SecureRowSums` step without the dropped client. Any later dropout will only result in that client not receiving labels for its data, but it will not affect the results for the other clients.

## 4 Experiments

In the following section we report experimental results for XCLP. We present two applications in the context of federated learning. In Section 4.1 we illustrate how XCLP can be used in a one-shot way to infer labels at prediction time. In Section 4.2 we show how XCLP can be used for federated semi-supervised learning by integrating it into a federated averaging training loop. As our emphasis here is on accuracy, not real-world efficiency, we use a simulated setting of federated learning, rather than physically distributing the clients

Table 2: XCLP for prediction (Fed-ISIC2019 dataset): Classification accuracy [in %] for leave-one-client-out experiments (average and standard deviation across three runs).

| left-out client | FedAvg | XCLP |
|---|---|---|
| Client 1 | $35.15 \pm 0.96$ | $\mathbf{38.89 \pm 1.34}$ |
| Client 2 | $\mathbf{68.75 \pm 0.72}$ | $67.05 \pm 0.54$ |
| Client 3 | $51.94 \pm 2.52$ | $\mathbf{62.02 \pm 1.13}$ |
| Client 4 | $42.72 \pm 0.79$ | $\mathbf{54.84 \pm 0.78}$ |
| Client 5 | $41.53 \pm 1.60$ | $\mathbf{52.28 \pm 0.58}$ |
| Client 6 | $46.92 \pm 3.69$ | $\mathbf{61.27 \pm 3.19}$ |

Table 3: XCLP for prediction (Fed-ISIC2019 dataset): Classification accuracy [in %] when different fractions, $\alpha$, of training data are labeled. FedAvg and *XCLP (labeled)* use only the labeled part of the training set, *XCLP (labeled+unlabeled)* uses also the unlabeled part.

| | FedAvg | XCLP (labeled) | XCLP (labeled+unlabeled) |
|---|---|---|---|
| $\alpha = 0.05$ | $43.78 \pm 0.88$ | $47.48 \pm 1.12$ | $\mathbf{49.50 \pm 1.42}$ |
| $\alpha = 0.1$ | $46.83 \pm 0.75$ | $52.34 \pm 0.93$ | $\mathbf{53.78 \pm 0.90}$ |
| $\alpha = 0.2$ | $52.87 \pm 0.14$ | $59.49 \pm 0.88$ | $\mathbf{61.09 \pm 0.66}$ |
| $\alpha = 1.0$ | $63.78 \pm 0.20$ | $\mathbf{74.22 \pm 0.67}$ | $\mathbf{74.22 \pm 0.67}$ |

across multiple devices. Therefore, we also use plaintext placeholders for the cryptographic steps that have identical output. Source code for our experiments can be found at https://github.com/jonnyascott/xclp.

## 4.1 XCLP for prediction

The most straightforward application of XCLP is as a method to predict labels for new data at inference time. For this setting suppose a set of clients, $P$, possess training data $X^{(j)} \in \mathcal{X}^{n^{(j)}}$ from some input space $\mathcal{X}$, and a (potentially partial) label matrix $Y^{(j)} \in \{0,1\}^{n^{(j)} \times C}$. The goal is to infer labels for new batches of data, $X_{\text{new}}^{(j)}$. Note that the above setting is general enough to encompass a number of settings, including clients with fully labeled or fully unlabeled training data. Also included is the possibility that a client $j$ has no training data to contribute, $X^{(j)} = \emptyset$, but has a batch of new data to be labeled, $X_{\text{new}}^{(j)} \neq \emptyset$.

By $h : \mathcal{X} \to \mathbb{R}^d$ we denote a preprocessing function, such as a feature extractor. Each client applies $h$ to all their data points, train and new, to obtain their input vectors to the XCLP routine, $V^{(j)} \coloneqq h\big(X^{(j)} \cup X_{\text{new}}^{(j)}\big)$. Running Algorithm 1 on $(V^{(j)}, Y^{(j)})_{j \in P}$, each client obtains $\hat{Y}^{(j)}$, which are label assignments for all of their data points, in particular including $X_{\text{new}}^{(j)}$ as desired.

**Experimental Setup** We use the Fed-ISIC2019 dataset (Ogier du Terrail et al., 2022), a real-world benchmark for federated classification of medical images. It consists of a total of 23247 images across 6 clients. The dataset is highly heterogeneous in terms of the amount of data per client, the classes present at each client as well as visual content of the images. As baseline classifier, we follow (Ogier du Terrail et al., 2022) and use an EfficientNet (Tan & Le, 2019), pretrained on ImageNet, which we finetune using federated averaging. As preprocessing functions, $h$, we use the feature extraction layers of the network either at the point of initialization (pretrained) or after the finetuning. Appendix A.1 gives full details of the experimental setup.

**Results** Table 1 reports results for the setting in which all clients contribute fully-labeled training data and have new data that should be classified. The left columns ("pretrained") illustrate the one-shot setting: no network training is required, only features are extracted once using a pretrained network, and XCLP is run once to infer labels. XCLP performs better than per-client label propagation here, except for the largest dataset size, indicating that in the limited data regime, it is indeed beneficial to build the data graph jointly from all data rather than separately on each clients. The other three columns ("finetuned") illustrate that

---

**Algorithm 2:** FedAvg+XCLP

---

**Input:** partially labeled training data $(X^{(j)}, Y^{(j)})_{j=1}^m$

1   $\theta = (\phi, \psi) \leftarrow$ `InitializeModelParameters`
2   **for** *round* $t \in [1, \ldots T]$ **do**
3     $P \leftarrow$ server randomly selects $\tau m$ clients
4     Server broadcasts $\theta$ to each client in $P$
5     **for** *client* $j \in P$ *in parallel* **do**
6       $V^{(j)} \leftarrow f_\phi(X^{(j)})$
7       $\hat{y}^{(j)}, \omega^{(j)} \leftarrow$ `XCLP`$(V^{(j)}, Y^{(j)}, P,$ Server$)$
8       $\theta^{(j)} \leftarrow$ `ClientUpdate`$(X^{(j)}, \hat{y}^{(j)}, \omega^{(j)}; \theta)$
9       Client $j$ sends $\theta^{(j)}$ to the server
10    $\theta \leftarrow$ `ServerUpdate`$\big((\theta^{(j)})_{j \in P}\big)$

**Output:** model parameters $\theta$

---

with a better –task-adapted– feature representation, XCLP still outperforms per-client LP, and also achieves better accuracy than predicting labels using only the network.

Table 2 reports results for a more challenging setting. We adopt a leave-one-client-out setup in which one client does not contribute labeled training but instead its data is meant to be classified. Given the heterogeneity of the clients, this means the classifiers have to overcome a substantial distribution shift. XCLP achieves better results than a network trained by federated averaging in all but one case, where in several cases the advantage is quite substantial. Note that per-client LP is not applicable here, as the new data is all located on a client that does not have labeled training data.

Finally, we also conduct on ablation study on the effect of unlabeled training data on XCLP, that is when $Y^{(j)}$ is a (strictly) partial label matrix. In this case the unlabeled training data does not contribute label information towards inference on $X_{\text{new}}^{(j)}$, but does contribute to a more densely sampled graph. Table 3 shows the results: the *FedAvg* column is identical to the one in Table 1, because the federated averaging training does not benefit from the additionally available unlabeled training data. Similarly, the result for *XCLP* with only labeled data are the same as for XCLP in Table 1. However, allowing XCLP to exploit the additional unlabeled data, however, indeed improves the accuracy further. This results once again shows the benefits of exploiting unlabeled data, especially when the amount of label data is small.

### 4.2   XCLP for federated semi-supervised learning

We now describe how XCLP can be applied iteratively during a training loop in the context of federated semi-supervised learning. *FedAvg+XCLP*, shown in pseudocode in Algorithm 2, follows a general FL template of alternating local and global model updates. As such, it is compatible with most existing FL optimization schemes, such as *FedAvg* (McMahan et al., 2017), *FedProx* (Li et al., 2020), or *SCAFFOLD* (Karimireddy et al., 2020). The choice of scheme determines the exact form of the `ClientUpdate` and `ServerUpdate` routines.

The first step (line 1) is to initialize the model parameters, $\theta = (\phi, \psi)$, where $f_\phi : \mathcal{X} \to \mathbb{R}^d$ is the feature extraction part of a neural network and $f_\psi : \mathbb{R}^d \to \mathbb{R}^C$ is the classifier head. The initialization could be random, using weights of a pretrained network, by an unsupervised technique, such as contrastive learning, or by a supervised step, such as federated training on only the labeled examples.

We then iterate the main training loop over $T$ rounds. To start each round the server samples some fraction $\tau$ of the $m$ total clients. These clients receive the current model parameters from the server (line 4) and embed their labeled and unlabeled data with the feature extractor, $f_\phi$ (line 6). Clients and server then collaboratively run XCLP on these feature vectors (line 7). As output of this step each client updates the pseudo-labels and confidence values for their unlabeled data, which they then use for local supervised training (line 8). Lastly, clients send the updated local models to the server (line 9) which aggregates them (line 10).

Table 4: XCLP for federated SSL: classification accuracy [in %] on federated CIFAR-10. $m$ is the number of clients, $m_L$ the number of clients with labeled data, $n_L$ is the total number of labels across all clients. i.i.d. and non-i.i.d. refer to how the data is split among the clients. For details, see the main text and Appendix A.2.

| | CIFAR-10, i.i.d. ($m = 100$) | | | |
| --- | --- | --- | --- | --- |
| | $m_L = 100$ | | $m_L = 50$ | |
| Method | $n_L = 1000$ | $n_L = 5000$ | $n_L = 1000$ | $n_L = 5000$ |
| FedAvg (labeled only) | $55.46 \pm 0.43$ | $76.13 \pm 0.46$ | $56.97 \pm 0.59$ | $80.36 \pm 0.07$ |
| FedAvg+perclientLP | $61.75 \pm 2.22$ | $85.11 \pm 0.73$ | $65.29 \pm 2.50$ | $84.41 \pm 0.25$ |
| FedAvg+network | $60.12 \pm 0.15$ | $79.45 \pm 0.31$ | $59.14 \pm 0.35$ | $81.04 \pm 0.20$ |
| FedMatch | $50.93 \pm 0.56$ | $72.22 \pm 0.14$ | $57.10 \pm 0.46$ | $77.80 \pm 0.32$ |
| FedSiam | $67.02 \pm 0.98$ | $82.06 \pm 0.56$ | $62.98 \pm 1.61$ | $78.45 \pm 0.34$ |
| FedSem+ | $59.98 \pm 0.49$ | $79.49 \pm 0.15$ | $59.67 \pm 0.47$ | $80.94 \pm 0.25$ |
| FedAvg+XCLP (ours) | $\mathbf{70.91 \pm 0.71}$ | $\mathbf{86.65 \pm 0.16}$ | $\mathbf{70.81 \pm 1.65}$ | $\mathbf{86.29 \pm 0.34}$ |
| | CIFAR-10, non-i.i.d. ($m = 100$) | | | |
| | $m_L = 100$ | | $m_L = 50$ | |
| Method | $n_L = 1000$ | $n_L = 5000$ | $n_L = 1000$ | $n_L = 5000$ |
| FedAvg (labeled only) | $50.94 \pm 0.14$ | $75.34 \pm 1.38$ | $53.26 \pm 0.69$ | $79.65 \pm 0.12$ |
| FedAvg+perclientLP | $50.94 \pm 0.14$ | $76.61 \pm 1.50$ | $53.26 \pm 0.69$ | $79.65 \pm 0.12$ |
| FedAvg+network | $60.60 \pm 0.60$ | $80.07 \pm 0.53$ | $59.82 \pm 1.05$ | $81.14 \pm 0.23$ |
| FedMatch | $50.71 \pm 1.57$ | $71.99 \pm 0.70$ | $48.24 \pm 0.86$ | $66.37 \pm 0.41$ |
| FedSiam | $67.85 \pm 0.26$ | $82.23 \pm 0.46$ | $62.29 \pm 1.84$ | $78.84 \pm 0.72$ |
| FedSem+ | $60.93 \pm 0.97$ | $79.70 \pm 0.78$ | $59.74 \pm 0.74$ | $81.30 \pm 0.09$ |
| FedAvg+XCLP (ours) | $\mathbf{73.76 \pm 0.71}$ | $\mathbf{85.53 \pm 0.56}$ | $\mathbf{70.01 \pm 1.29}$ | $\mathbf{85.42 \pm 0.43}$ |

Table 5: XCLP for federated SSL: classification accuracy [in %] on federated CIFAR-100 and Mini-ImageNet. $m$ is the number of clients, $m_L$ the number of clients with labeled data, $n_L$ is the total number of labels across all clients.

| | CIFAR-100, i.i.d. | | Mini-ImageNet, i.i.d. | |
| --- | --- | --- | --- | --- |
| | $m = m_L = 50$ | $m = m_L = 100$ | $m = m_L = 50$ | $m = m_L = 100$ |
| Method | $n_L = 5000$ | $n_L = 10000$ | $n_L = 5000$ | $n_L = 10000$ |
| FedAvg (labeled only) | $43.80 \pm 0.19$ | $53.91 \pm 0.25$ | $23.39 \pm 0.52$ | $31.72 \pm 0.54$ |
| FedAvg+network | $43.80 \pm 0.19$ | $54.19 \pm 0.21$ | $23.98 \pm 0.36$ | $31.86 \pm 0.57$ |
| FedAvg+perclientLP | $43.82 \pm 0.59$ | $54.38 \pm 0.36$ | $25.53 \pm 0.22$ | $33.09 \pm 0.62$ |
| FedAvg+XCLP (ours) | $\mathbf{50.19 \pm 0.60}$ | $\mathbf{57.00 \pm 0.08}$ | $\mathbf{26.93 \pm 0.41}$ | $\mathbf{35.78 \pm 0.56}$ |

The motivation for this approach comes from the insight gained in Section 4.1, that XCLP assigns high quality labels when run on features obtained from a trained network. Crucially, pseudo-labels assigned by XCLP are always recomputed when a client is sampled. Thus as the network features improve so too does the quality of the pseudo-labeling.

**Experimental Setup** We evaluate the accuracy of FedAvg+XCLP against other methods for federated SSL as well as report on ablation studies. We adopt a standard federated averaging scheme, in which `ClientUpdate` consists of running 5 epochs of SGD with confidence-weighted cross-entropy loss on the local device and `ServerUpdate` simply averages the local client models.

We use three standard datasets: CIFAR-10 (Krizhevsky, 2009), which has 10 classes and is used in several previous federated SSL works (Jeong et al., 2021; Long et al., 2020; Albaseer et al., 2020), as well as the more difficult CIFAR-100 (Krizhevsky, 2009) and Mini-ImageNet (Vinyals et al., 2016) which have 100 classes. To the best of our knowledge ours is the first work in this federated SSL setting to evaluate on these more challenging datasets. The datasets are split in different ways (different number of clients, different amounts of labeled data, i.i.d. vs non-i.i.d.) to simulate a diverse range of federated settings.

We compare FedAvg+XCLP to a broad range of other methods. To enable a fair comparison of results, all methods use the same network architecture, and hyper-parameters are chosen individually to maximize each method's performance. From the existing federated SSL literature, we report results for *FedMatch*

(Jeong et al., 2021), *FedSiam* (Long et al., 2020) and *FedSem+*, which follows (Albaseer et al., 2020) but additionally uses confidence-based sample weights, as we found these to consistently improve its accuracy. Additional baselines are two methods that follow the same structure as Algorithm 2 but use alternative ways to obtain pseudo-labels: from per-client label propagation (FedAvg+perclientLP) or from the network's classifier predictions (FedAvg+network). Finally, we also report results for training in a supervised manner on only the available labeled examples, *FedAvg (labeled only)*. Note that we do not include comparisons to Zhang et al. (2021b) and Diao et al. (2022) as these methods address a different federated SSL scenario in which the server has access to labeled data while the clients have no labels.

**Results**   We report the results of our experiments in Tables 4 and 5 as the average accuracy and standard deviation over three random splits of the data for each setting. Table 4 provides a comparison of XCLP to other approaches and baselines in the standard setting of CIFAR-10 with 100 clients, which has been used in prior work. In each case, we report results when 1,000 or 5,000 of the data points are labeled. Either all or half of the clients have labels, with classes distributed either i.i.d. or non-i.i.d. across clients. Table 5 reports on the harder situation with many more classes, which prior work has not attempted. Across the board, XCLP achieves the best results among all methods, while of the other methods, none has a consistent advantage over the others. In addition to these general observations the results offer a number of more specific insights.

Firstly, in nearly all cases semi-supervised methods outperformed the labeled only FedAvg baseline. This is to be expected given the additional (unlabeled) data available to the SSL methods. A notable exception to this, however, is FedAvg + perclientLP in the non-i.i.d. scenario. In this case in three of the four cases perclient-LP actually degraded the performance of the initial supervised model, which caused automatic model selection to deactivate it. A likely reason for this poor performance is the presence of classes in the client test data which do not appear in the labeled portion of the clients training data. In this situation perclient-LP is not able to predict for these classes on the test data. In contrast XCLP is unlikely to be affected by this issue, as more clients are pooled together and hence more classes are present in the labeled training data. This is reflected in the strong performance of FedAvg + XCLP in the non-i.i.d. setting.

Secondly, we observe that the biggest gains from incorporating XCLP occur in the regime where little labeled data is available. In particular we see that with ample labeled data available per client, in the i.i.d. settings, perclientLP performs not so much worse than XCLP. This is to be expected as when each client already has ample labels available then the potential gain of collaboration with other clients is of course lower.

## 5   Conclusions and Limitations

In this work we introduced XCLP, a method for transductively predicting labels for unlabeled data points in a federated setting, where the data is distributed across multiple clients. It makes use of cryptographic routines to preserve data confidentiality when estimating a joint neighborhood graph over the data and propagates label information across this graph by distributing the computation among the clients. We presented two applications of XCLP, inferring labels for new (test) data, and training on partly labeled data in a federated semi-supervised setting. In our experiments XCLP led to substantial improvements in classification accuracy in both applications, especially in the most challenging (but often realistic) setting when the amount of labeled data per client is limited.

XCLP ensures that a client's data remains anonymous, in the sense that it is not directly exposed to or shared with any other party, a notion of privacy we referred to as data confidentiality. This is achieved through the guarantees provided by the cyrryptographic subroutines. While data confidentiality is a fundamental building block of private machine learning paradigms, such as federated learning, on its own it does not guarantee that nothing can be learned about the clients or their data. Indeed it is common practice in private machine learning to combine data confidentiality with other notions of privacy, such as differential privacy, in order to obtain stronger privacy guarantees. Extending XCLP to integrate such notions of privacy is an interesting direction for future work.

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

# A  Experimental Details

## A.1  XCLP for Prediction

**Dataset**  We use the Fed-ISIC2019 (Ogier du Terrail et al., 2022) dataset which contains over 20, 000 images of skin lesions. The task is to predict melanoma (cancer) types. There are 6 clients, naturally defined by hospital and scanner used. As a result the data of each client is highly heterogeneous in terms of the amount of data per client (12413, 3954, 3363, 2259, 819, 439 examples for each client respectively), the classes present at each client as well as visual content of the images. Due to the class imbalance in the dataset the evaluation metric used is balanced accuracy.

**Network**  Following Ogier du Terrail et al. (2022) we use an EfficientNet (Tan & Le, 2019) pretrained on ImageNet, which we denote by $f$. We initialize a new final linear layer and fine-tune the whole network using federated averaging as described in Ogier du Terrail et al. (2022).

**Hyper-parameters**  We set all hyper-parameters for `FederatedAveraging` to the values specified in Ogier du Terrail et al. (2022) except we increase the number of training rounds to $T = 40$ as we found that the accuracy to improve with further training. Parameters for XCLP (LSH dimension, $k$-NN parameter) are chosen using cross-validation. We use $L = 1024$ and $k = 3$. We fix the parameter $\alpha = 0.99$.

## A.2  XCLP for federated semi-supervised learning

**Datasets**  We evaluate XCLP on three standard datasets for multi-class classification: CIFAR-10 (Krizhevsky, 2009), which has 10 classes and is used in previous federated SSL works, as well as the more difficult CIFAR-100 (Krizhevsky, 2009) and Mini-ImageNet (Vinyals et al., 2016) which both have 100 classes. To the best of our knowledge ours is the first work in this federated SSL setting to evaluate on these more challenging datasets. All three datasets consist of 60,000 images which we split into training sets of size $n \coloneqq 50{,}000$ and test sets of size 10,000. From the training set, $n_L$ examples are labeled and the remaining $n - n_L$ are unlabeled. For CIFAR-10 we evaluate with $n_L = 1{,}000$ and 5,000. For CIFAR-100 and Mini-ImageNet we take $n_L = 5{,}000$ and 10,000.

**Federated Setup**  We simulate a FL scenario by splitting the training data (labeled and unlabeled) between $m$ clients. $m_L$ of these have partly labeled data, while the others have only unlabeled data. Each client is assigned a total of $n/m$ data points of which $n_L/m_L$ are labeled if the client is one of the $m_L$ which possess labels. We simulate statistical heterogeneity among the clients by controlling the number of classes each client has access to. In the i.i.d. setting all clients have uniform class distributions and receive an equal number of labels of each class. In the non-i.i.d. setting we assign a class distribution to each client and clients receive labels according to their own distribution.

**Networks**  Following prior work, we use 13-layer CNNs (Tarvainen & Valpola, 2017) for CIFAR-10 and 100 and a ResNet-18 (He et al., 2016) for Mini-ImageNet. Feature extractors are all layers except the last fully connected one, thus embeddings have dimension 128 and 512, respectively.

**Hyper-parameters**  We choose hyper-parameters for all methods based on training progress (LSH dimension, $k$-NN parameter) or accuracy on a held-out validation set consisting of 10% of the training data (batch size, learning rate).

**Federated learning parameters**  We set the number of clients to $m = 100$, except for our experiments on CIFAR-100 and Mini-ImageNet with $n_L = 5000$. In these cases we set $m = 50$ as it is not possible to create an i.i.d. split of the data over 100 clients since the number of classes (C=100) is too large. For CIFAR-10 we set the number of clients which possess labels to $m_L = 100$ and $m_L = 50$. On CIFAR-100 and Mini-ImageNet we set $m_L = m$.

The `ClientUpdate` step corresponds to $E$ epochs of stochastic gradient descent (SGD) of a loss function. We set the number local epochs to $E = 5$ and the loss function is (per sample weighted) cross-entropy loss. The

`ServerUpdate` step corresponds to averaging the model updates:

$$\texttt{ServerUpdate}(\theta^{(j)} \text{ for } j \in P) = \frac{1}{|P|} \sum_{j \in P} \theta^{(j)}.$$

The number of training rounds is set to $T = 1500$ and the number of clients sampled by the server per training round is set to 5, so $\tau = 0.05$ when $m = 100$ and $\tau = 0.1$ when $m = 50$. Note that when $m_L < m$ we ensure that the server samples $\tau m_L$ clients from the labeled portion (and $\tau(m - m_L)$ from the unlabeled) to ensure that there are some labels present in the graph.

**Network training parameters** We use standard data augmentation following Tarvainen & Valpola (2017). On CIFAR-10 and CIFAR-100 this is performed by 4Œ4 random translations followed by a random horizontal flip. On Mini-ImageNet, each image is randomly rotated by 10 degrees before a random horizontal flip. We use weight decay for all network parameters which is set to $2 \times 10^{-4}$. When carrying out SGD in the `ClientUpdate` we use batches of data $B = B_L \cup B_U$ where $B_L$ is a batch of labeled data and $B_U$ is a batch of pseudo-labeled (previously unlabeled) data. We set $|B_L|$ according to how many labeled samples the client has available, $|B_L| = \min(50, \#labels)$. We set $|B_U| = |B_L|$. Learning rate for SGD is set according to this batch size. On CIFAR-10, for $|B_L| < 50$ we set the learning rate to 0.1 and for $|B_L| = 50$ we set the learning rate to 0.3. On CIFAR-100 and Mini-ImageNet we always have $|B_L| = 50$ and we set the learning rates to 0.5 and 1.0 respectively. We decay the learning rate using cosine annealing so that the learning rate would be 0 after 2000 rounds.

**XCLP parameters** We set the LSH dimension to $L = 4096$ as this gave near exact approximation of the cosine similarities while still being computationally fast (less than 1 second per round). We set the sparsification parameter to $k = 10$, so that each point is connected to its 10 most similar neighbors in the graph, and the label propagation parameter to $\alpha = 0.99$.

# B Additional Experiments

## B.1 XCLP for Prediction

We include extra experiments to test the performance of XCLP for prediction on an additional dataset. We follow the notation and setup detailed in Section 4.1.

**Dataset** We use the FEMNIST Caldas et al. (2018) dataset. FEMNIST is a federated dataset for handwritten character recognition, which has 62 classes (digits and lower/upper case letters). It has 817,851 samples and we keep the natural partition into 3597 clients based on the writer that wrote each character. The clients are non-i.i.d., as they heterogeneous in the amount of data they possess, the classes they have, as well as the data distributions themselves.

**Experimental Setup** We consider a setting where each client possesses partly labeled training data and wishes to infer labels for their own unlabeled new data. We looks at a range of different labeling scenarios based on what fraction, $\alpha$, of each client's training data is labeled. Furthermore, due to the large number of clients and datapoints in FEMNIST we reduce communication overhead by partitioning the clients into large groups (we use 50 groups, each with approximately 700 clients) and run XCLP separately on each groups of clients. Per-client LP remains as described in 4.1.

We investigate two different choices for the preprocessing function $h$. For the first we simply use the identity (i.e. no embedding) and run both per-client LP and XCLP directly on the raw data. For the second we use a linear embedding that we trained over the client data. Specifically, following the low training overhead approach of 4.1, we train a two layer linear MLP using Federated Averaging and use the first layer linear embedding as our preprocessing function.

**Results** Table 6 reports the results obtained when different fractions $\alpha$ of each client's training data are labeled. The left columns ("identity") give results for the one-shot setting, where no training is required,

Table 6: XCLP for prediction (FEMNIST dataset): Classification accuracy [in %] with two different preprocessing functions (identity and linear embedding) and different fractions of labeled data available (average and standard deviation across three runs).

| | identity | | linear | | |
|---|---|---|---|---|---|
| *Fraction of labeled data* | *per-client LP* | *XCLP* | *FedAvg* | *per-client LP* | *XCLP* |
| $\alpha = 0.1$ | $33.74 \pm 0.20$ | $\mathbf{49.29 \pm 0.16}$ | $\mathbf{53.09 \pm 0.19}$ | $37.27 \pm 0.12$ | $52.39 \pm 0.30$ |
| $\alpha = 0.2$ | $44.79 \pm 0.03$ | $\mathbf{53.93 \pm 0.04}$ | $57.54 \pm 0.46$ | $50.89 \pm 0.11$ | $\mathbf{58.57 \pm 0.13}$ |
| $\alpha = 0.5$ | $50.99 \pm 0.06$ | $\mathbf{59.37 \pm 0.13}$ | $61.07 \pm 0.49$ | $59.20 \pm 0.19$ | $\mathbf{63.46 \pm 0.11}$ |
| $\alpha = 1.0$ | $51.91 \pm 0.07$ | $\mathbf{62.35 \pm 0.07}$ | $62.90 \pm 0.31$ | $59.42 \pm 0.11$ | $\mathbf{66.18 \pm 0.20}$ |

and LP is run directly on the client features. The right columns ("linear") illustrate that by embedding the features using a simple linear layer we are able to improve the performance of XCLP. In all cases XCLP substantially outperforms per-client LP. The difference is even more noticeable than in 4.1, presumably due to the smaller amount of data present at each client.

## C  Additional Background

### C.1  Propagating values along a data graph

For a graph with $n$ vertices and adjacency matrix $\mathcal{W} \in \mathbb{R}^{n \times n}$ of edge weights, a vector of values, $y \in \mathbb{R}^n$, can be propagated to neighboring vertices by forming $z = \mathcal{W}y$. By repeatedly multiplying with $\mathcal{W}$ values can be propagated all along the graph structure (Zhu, 2005).

In the context of propagating labels, one wants not only to propagate the labels to unlabeled points, but also to prevent the information at labeled points to be forgotten. For that, one can uses an extended update rule

$$z_{t+1} = \alpha \mathcal{W} z_t + y \tag{1}$$

where $\alpha \in (0, 1)$ is a trade-off hyperparameter. For $\|\mathcal{W}\| < 1/\alpha$, this process has the closed form expression

$$z_\infty = (\mathrm{Id} - \alpha \mathcal{W})^{-1} y, \tag{2}$$

as its $(t \to \infty)$-limit. This can be seen from the fact that Equation 1 is a contraction with $z_\infty$ as fixed point.

In Section 3.1 we make use of this fact together with the observation that Equation 2 can readily be applied to vectors-valued data with a matrix $Y$ in place of $y$. However, the propagation step will not preserve normalization, e.g. of the $L^1$-norm. When such a property is required, e.g. for the calculation of entropy-based confidence, normalization has to be performed explicitly post-hoc.

### C.2  Computing similarity from hashed data

Locality-sensitive hashing (LSH) (Indyk & Motwani, 1998) is a procedure for hashing real-valued vectors into binary vectors while preserving their pairwise similarity. Let $v \in \mathbb{R}^d$ be a vector. To encode $v$ into a binary vector $b$ of length $L$, LSH randomly samples $L$ hyperplanes in $\mathbb{R}^d$. For each hyperplane it checks whether $v$ lies above or below it and sets the $i$th bit in $b$ as 1 or 0 accordingly. Formally, $b_i = \mathbb{1}_{\langle v, u_i \rangle \geq 0}$, where $u_i \in \mathbb{R}^d$ is the normal vector of the $i$th hyperplane. A key property of LSH is that it approximately preserves cosine-similarity. Concretely, for vectors $v_1, v_2$ with LSH encodings $b_1, b_2$ (compute with the same projections), one has

$$\frac{\langle v_1, v_2 \rangle}{\|v_1\| \|v_2\|} \approx \cos(\pi h(b_1, b_2)/L) \tag{3}$$

where $h$ is the Hamming distance (number of bits that differ) between two binary vectors. The reason is that the probability of $b_1$ and $b_2$ differing at any bit $i$ is the probability that the $i$-th sampled hyperplane lies between $v_1$ and $v_2$, which is equal to $\angle(v_1, v_2)/\pi$. By the law of large numbers, the more hyperplanes one samples, the better the approximation quality.

