# OpenReview forum: "Cross-client Label Propagation for Transductive and Semi-Supervised Federated Learning"
_TMLR — Accepted by TMLR_

### Review · Reviewer_Dy9w · 2023-07-21

**Summary Of Contributions:**

The paper introduces a novel method for transductive and semi-supervised federated learning to estimate the labels of unlabeled data points across all clients without sharing the data. For this, the proposed method, Cross Client Label Propagation (XCLP), estimates a data graph jointly from the clients’ data and computes the labels by propagating label information across the graph. Additionally, XCLP uses two cryptographically secure protocols: secure Hamming distance and secure summation.
The authors show the benefits of their proposed approach in different federated learning scenarios, including one-shot label prediction and semi-supervised learning. On one side, the results show that the computation of the graph using multiple clients produces significant improvements with respect to strategies based on the propagation of labels individually for each client. In semi-supervised federated learning, XCLP outperforms other state-of-the-art methods.


**Audience:**

Yes

**Broader Impact Concerns:**

No concerns.

**Claims And Evidence:**

Yes

**Requested Changes:**

+ Evaluation of XCLP for label inference in at least another dataset.
+ Extend the discussion and analysis of the results in Section 4.2.
+ Empirical analysis of the computational complexity (XCLP vs per-client LP).


**Strengths And Weaknesses:**

Strengths:
+ Very interesting approach for the computation of the joint graph for propagating the information of the labels. Section 3 is well structured and explained and includes the analysis of different properties of the algorithm, such as efficacy, privacy, efficiency and robustness.
+ Interesting use of the two secure protocols for keeping the privacy of the clients’ data for propagating the labels.
+ Experimental evaluation seems convincing and the uses cases illustrate well the capabilities of XCLP and its benefits compared to other state-of-the-art methods, including cases like semi-supervised learning and label inference in one-shot (although there are some improvements that can be addressed in the experiments).

Weaknesses:
- In the experiments, for label inference (Section 4.1), only one dataset is considered in the experiments (Fed-ISIC 2019). Although the analysis of this dataset is complete and includes the ablation study in Table 3, it would be necessary to confirm the results with at least another dataset.
- The discussion of the results on semi-supervised learning in Section 4.2 is too brief and shallow (just one small paragraph).
- The computational complexity of the method is not analyzed in the experiments. For example, it would be interesting to compare the computational cost of per-client LP and XCLP in Section 4.1.

---

> ### Author Response · Authors · 2023-08-19
> **Response to Reviewer Dy9w**
>
> Thank you for your feedback. We address your specific questions below.
>
> > In the experiments, for label inference (Section 4.1), only one dataset is considered in the experiments (Fed-ISIC 2019). Although the analysis of this dataset is complete and includes the ablation study in Table 3, it would be necessary to confirm the results with at least another dataset.
>
> Thank you for the suggestion. We have added extra experiments on a new dataset for label inference. We have now also included results when running label inference on the FEMNIST federated dataset. Full details on the dataset, experimental setup and results can be found in the updated manuscript in Appendix B.
>
> > The discussion of the results on semi-supervised learning in Section 4.2 is too brief and shallow (just one small paragraph).
>
> Following your suggestion we have now extended the discussion to include a more detailed analysis of the results, please see Section 4.2 of the updated manuscript.
>
> > The computational complexity of the method is not analyzed in the experiments. For example, it would be interesting to compare the computational cost of per-client LP and XCLP in Section 4.1.
>
> XCLP is, of course, computationally more expensive than per-client LP. On the one hand there is the overhead of the cryptographic steps, and on the other hand it performs operations on larger matrices. In terms of empirical values, we had considered measuring actual runtimes, but ultimately came to a conclusion that it would not provide a fair or insightful comparison. The reason is that cryptographic protocols need to be implemented professionally, in particular making use of proper CPU instructions, only then do they run efficiently. Otherwise, XCLP’s runtime would be completely dominated by the overhead of a naive implementation (imagine, e.g., having to call an external library to process each individual bit from python rather than executing a single 512-bit AVX instruction).
>
> In terms of numeric complexity, an analysis is possible, and have added it to Section 3.3.
> Assume batches of $p$ clients, each with $n$ data points, $m$ of which are labeled. The input dimension is $d$ and the number of labels $C$. For per-client LP, each client has to compute $n^2/2$ $d$-dimensional inner products, invert the resulting $n\times n$ matrix, and multiply a sub-matrix of size $m\times n$ it with a label matrix of size $m \times C$. The complexity per client is $O(dn^2+n^3+Cmn)$. This is also the total complexity, if all clients can operate in parallel. In practice, we expect $d>n>C$, so the dominant term is $O(dn^2)$. The server does not contribute any computation.
>
> For XCLP with $L$-dimensional bit vectors, each client computes an $n\times L$ binary data matrix, which has complexity $O(dnL)$. Then, the clients jointly compute all $p^2n^2/2$  pairwise Hamming distances, which requires $O(Lp^2n^2)$ operations and has complexity $O(Lpn^2)$ if run in parallel by the clients. The server inverts the matrix at cost $O(p^3n^3)$. Then, each client multiplies an $pn\times m$ sized part of the resulting matrix with their $m\times C$ label matrix, which costs $O(Cpnm)$ per client, and also overall complexity $O(Cpnm)$ if run on the clients in parallel. The most costly step is the matrix inversion, but that is done on the server, so in practice it can be considered almost free. Assuming $L>C$ and $np>d$, the clients’ costs are dominated by the $O(Lpn^2)$ term. In total, with a typical trade-off of, e.g. $L=8d$, XCLP requires $8p$ times more numeric operations than local LP.
> Note, however, that on an absolute scale, all of these values are rather small. For $p=10$,  $d=512$, $L=4096$, the number of numeric operations a client has to perform per datapoint is  in the order of $10^7$, which is fewer than the operations needed to compute the data point’s feature representation using even a small neural network. Of course, this argument does not take into account the cryptographic aspect, which will cause a constant slowdown, see above.
> Note also, that one has good control over the total cost, as $L$ and $p$ (and potentially $d$) are design choices.

---

### Review · Reviewer_8NrT · 2023-07-22

**Summary Of Contributions:**

This paper presents a new method for transductive and semi-supervised federated learning called Cross-Client Label Propagation (XCLP). XCLP estimates a data graph from multiple clients' data and computes labels for unlabeled data by propagating label information across the graph. To maintain privacy, XCLP uses two cryptographically secure protocols: secure Hamming distance computation and secure summation.

The paper makes several contributions to the field of federated learning:
1. A new federated version of label propagation algorithm is proposed. XCLP allows for multiple clients to jointly infer labels from a cross-client data graph without them having to share their data.
2. Two new cryptographically secure protocols are used in XCLP: secure Hamming distance computation and secure summation. These protocols are used to maintain privacy in XCLP, as they avoid clients having to share their data with anyone.


**Audience:**

Yes

**Broader Impact Concerns:**

The reviewer does not find any concerns related to the ethical implications.

**Claims And Evidence:**

Yes

**Requested Changes:**

The reviewer does like this work and enjoy reading the paper even though there are some potential improvements about it. The proposed adjustments are listed as follows. The reviewer knows these changes may need much work and some suggestions are beyond the scope of this paper. So it is ok if the authors find it too hard to make these changes. The reviewer recommends for the acceptance of this paper based on its current status, but these changes could make this work even better. **All the following changes are not critical but could strengthen the work.** More details can be found in Strengths And Weaknesses section.

1. More theoretical analysis on the proposed method on its accuracy guarantee compared to the original naive version.
2. More recent baselines comparison for the federated semi-supervised learning case setting.

**Strengths And Weaknesses:**

Strengths:
1. The major contribution of work is quite significant to the federated community and semi-supervised learning community. Because it proposes a novel distributed label propagation paradigm under the federated learning context. It will be beneficial to researchers from both communities.

2. The proposed method is well-motivated with clear logics. The main challenge of extending label propagation algorithm is how to construct a similarity graph without any data leakage and how to do label inference without privacy concerns. The proposed method solves them nicely with strong motivations and reasonable techniques. For the graph construction, we do not compute the cos similarity directly on the data features from each client but on top the  Hamming matrix of LSH projection. Similarly, we do not do label inference directly on the whole similarity matrix in the server, but only distributes the corresponding sub-matrix to each client.

3. The writing is clear and easy to understand in general with necessary annotations.


Weaknesses:
1. The work is not quite theoretically sound. The reviewer does like the proposed method a lot, but it would be more interesting if the authors can provide some theoretical analysis of the proposed method. In fact, there are several potential aspects of doing so. For example, when compared to the classic centralized way of graph construction, how different would be the graph generated by the proposed method in terms of some distance metric? Are there any rigorous privacy guarantee about the two proposed secure protocols and will the performance of the label propagation be degraded compared to the naive version without privacy considerations?

2. Some more recent federated semi-supervised learning models should be added in Section 4.2. The current baselines are a bit of dated. In fact there are several new baselines[1][2] related to federated semi-supervised learning.

[1] SemiFL: Semi-Supervised Federated Learning for Unlabeled Clients with Alternate Training. In NeurIPS 2022.

[2] Towards Unbiased Training in Federated Open-world Semi-supervised Learning. In ICML 2023.

---

> ### Author Response · Authors · 2023-08-19
> **Response to Reviewer 8NrT**
>
> Thank you for your feedback and positive comments. We address your specific points below
>
> > The reviewer does like the proposed method a lot, but it would be more interesting if the authors can provide some theoretical analysis of the proposed method. [...]
>
> > When compared to the classic centralized way of graph construction, how different would be the graph generated by the proposed method in terms of some distance metric?
>
> This appears to be a misunderstanding. **The neighborhood graph computed by XCLP is identical to the one that would have been computed centrally** if both use same similarity function (i.e. Hamming distance). The Hamming construction is known to yield an $\epsilon$-approximation of the exact cosine distance, where the approximation quality is somewhat involved, but analyzed in [LSH]. Note it is in general not possible to theoretically quantify how different the neighborhood graphs themselves are going to be, because the k-NN process is discontinuous. In practice, however, we found no noticeable difference between the quality of LP with exact and approximate distance function.
> We will make these aspects clearer in the Efficacy part of Section 3.3.
>
> > Are there any rigorous privacy guarantees about the two proposed secure protocols.
>
> **The two secure protocols indeed come with privacy guarantees that they inherit directly from the cryptographic primitives they are based on** (see Section 3.2). *SecureHamming*, both based on oblivious transfer (SHADE) and based on partially homomorphic encryption (PHE), ensures that the server will learn only the entries of the Hamming matrix and nothing else. The clients will not learn anything about each others’ data. Similarly, *SecureRowSum* inherits the guarantees of multi-party-computation: the server receives only the sum of the per-client contributions, while the clients learn nothing about each other.
>
> > performance of the label propagation be degraded compared to the naive version without privacy considerations?
>
> See above. The graph produced with cryptographic security is identical to the one if the Hamming distances are computed naively. The approximation to the cosine distance obtained from the Hamming distance is slightly different from the exact one (controllable by $L$), but we observed no quality degradation.
>
> > More recent baselines comparison for the federated semi-supervised learning case setting.
>
> [1] SemiFL addresses an alternative federated semi-supervised scenario to the one we consider. In their setting the server has access to labeled data, while the clients have only unlabeled data. Their method cannot be applied to the case where the server has no data and likewise our method (for privacy reasons) should not be used in the case that the server contributes data to the learning.
>
> [2] is very recent. Thank you for bringing this to our attention. This method appears applicable in the setting we study. As we did not find publicly available code for it, we were not able to run experiments for this method in our setting. We will aim to do so for a revision of our work. Note, however, that [2] comes with rather different privacy guarantees, relying on the clients clustering their data and sending centroids to the server.

---

> > ### Comment · Reviewer_8NrT · 2023-08-19
> >
> > Thanks for your response. Many of my concerns are addressed. I am ok with the current version.

---

### Review · Reviewer_2fsr · 2023-08-01

**Summary Of Contributions:**

The paper studies a specific problem in the domain of semi-supervised federated learning. They call it {\em cross-client label propogation} (XCLP). The technique allows them to generate a data graph in a decentralized manner using data from the client at the node of this network. The client contains both labeled and unlabeled data.

**Audience:**

Yes

**Claims And Evidence:**

Yes

**Requested Changes:**

I can see that it would be difficult to give a theoretical guarantee for general graphs, but can the authors give some guarantee for special classes? I would suggest the following graphs as the starting point:
1. Line graph (or ring graph). Perhaps the most basic one.
2. Any d-regular graph.
3. Small cluster graph with each cluster highly connected. Most general graphs can be decomposed in the form of such clusters with very few edges between the nodes in two different clusters. It is known as sparse-dense decomposition (SDD).
4. Graph with high conductance. I would expect there would be some relationship between their algorithm and the expansion of the graph.

I do not see many other applications of the subroutines that the author proposed. This definitely can be my lack of understanding. Can the authors provide more motivation where these subroutines can be used? Currently, these subroutines seem more like a heuristic that can be used only to solve this problem.

**Strengths And Weaknesses:**

Strength:
- The problem statement is very nice and clean. It also has some applications in practice.
- The setting is interesting and one expects it in practice. Also, the prediction accuracy of unlabeled increases as we have access to more data. This is something expected and the paper passes this test.
- There is enough empirical evidence to support the main claims of the paper.

Weakness:
- I do not understand any novelty in the paper. The cryptographic techniques used in this paper are pretty basic and the variation of secure sum in itself is not necessarily that exciting to me.
- The work is more aligned with empirical evaluation.

---

> ### Author Response · Authors · 2023-08-19
> **Response to Reviewer 2fsr**
>
> Thank you for your feedback. We address the specific concerns individually below.
>
> > I do not understand any novelty in the paper. The cryptographic techniques used in this paper are pretty basic and the variation of secure sum in itself is not necessarily that exciting to me.
>
> The novelty in the paper is twofold. Firstly, the XCLP algorithm is the first federated learning method that is able to diffuse labels in a transductive manner across clients without them having to share their data. Secondly, the proposed applications of XCLP are new, namely stand-alone inference and as an integrated part of federated averaging in order to tackle the challenge of federated semi-supervised learning. The subroutines we propose to do the necessary federated computations, namely SecureHammingDistance and SecureRowSum, are also new and can be applied in a range of other settings which we discuss in more detail below. The cryptographic protocols we describe are tools to achieve these goals,  but we do not claim them to be major contributions by themselves.
>
> > I can see that it would be difficult to give a theoretical guarantee for general graphs, but can the authors give some guarantee for special classes?
>
> Could you clarify more precisely what you mean by *theoretical guarantees*? Note that our analysis in section 3.3 under Efficacy, shows that XCLP computes label propagation precisely as if the data were centrally available and we simply ran ordinary LP on it. As such, all classical theoretical results for LP, e.g. in [1,2] also hold for XCLP.
> So a study of what exactly XCLP learns for different graph structures is really a study of  theoretical guarantees of centralized LP which is interesting, but beyond the scope of this work.
>
> One related point to note is that in the case that client data is highly heterogeneous, in the sense that each client's data points share high similarity only with other points belonging to the same client, then the inferred graph computed by XCLP is the same as the one computed by local LP (#3 in your list of graph specialities) with non-interconnected clusters for each client. So in particular XCLP should not do worse than local LP, even when faced with high heterogeneity, which is when one would expect local LP to perform best.
>
> [1] X. Zhu, Z. Ghahramani. “Learning from labeled and unlabeled data with label propagation.” Technical Report CMU-CALD-02-107, 2002.
> [2] D. Zhou, O. Bousquet, T. Lal, J. Weston, B. Scholkopf. “Learning with local and global consistency”. In NeurIPS, 2003.
>
> > I do not see many other applications of the subroutines that the author proposed. This definitely can be my lack of understanding. Can the authors provide more motivation where these subroutines can be used?
>
> W envision a number of other potential applications for our routines.
> 1. The subroutine for privately computing the Hamming matrix H (line 5 of Algorithm 1) allows for the computation of pairwise similarity matrices over the data of all participating clients. We exploit this for the cosine distance (line 6), but other similarities are possible by employing other hashing schemes [RF]. Such pairwise similarity matrices lie at the cornerstone of many classical and modern machine learning methods, let it be nearest neighbor search [H], kernel methods [LK], such as Gaussian processes [GP], or even transformer attention layers [A]. Therefore we see several natural follow ups to our work exploring such methods in a federated setting.
> 2. The SecureRowSums (lines 11-14) essentially solves a specific form of distributed matrix multiplication. One matrix lies on the server, the other matrix is split row-wise between the clients. In the process, it is ensured that the clients receive only the rows of the solution that are relevant to them. This is in fact quite a general setting for a federated computation. Consider for instance solving the closed form of distributed linear least squares: $w=(X^T X)^{-1}Xy$. The matrix $X^TX$ is an outer product and can be computed on the server by summing the individual outer products of each client (also privately, by employing multi-party computation). The rest of the computation fits exactly our proposed blueprint: the server inverts the outer product matrix, and then wishes to compute a product with matrices that are split row-wise among the clients (Xy) and have the solutions be distributed row-wise among the clients.
>
> [H] J. Wang, H. T. Shen, J. Song, J. Ji. “Hashing for Similarity Search: A Survey”, https://arxiv.org/abs/1408.2927]
>
> [RF] Ali Rahimi, Benjamin Recht. “Random Features for Large-Scale Kernel Machines”, NeurIPS 2007.
>
> [AD] G. Pang, C. Shen, L. Cao, A. Van Den Hengel. “Deep Learning for Anomaly Detection: A Review”, ACM Computing Surveys 2021.
>
> [LK] B. Schölkopf, A. Smola. “Learning with Kernels”. MIT Press 2002.
>
> [GP] C. Rasmussen, C. Williams. “Gaussian Processes for Machine Learning”, MIT Press 2006
>
> [A] A. Vaswani. “Attention Is All You Need”, NeurIPS 2017.

---

### Author Response · Authors · 2023-10-30
**Camera Ready Revision**

Dear Reviewers and Editor,

Many thanks for your reviews and feedback on our submission. The camera ready revision has now been uploaded.

Best wishes,
The Authors

---

> ### Comment · Action_Editors · 2023-11-01
>
> Hi,
>
> As noted in the meta review, please include a short discussion about the type of privacy that the proposed approach achieves or does not achieve.
> This will be helpful for readers to properly place this paper's results in the context or other work in the literature.
>
> Thanks!

---

> > ### Author Response · Authors · 2023-11-18
> > **Camera Ready Reuploaded**
> >
> > Hi,
> >
> > Thanks for your feedback. We have now updated section 5, conclusion and limitations, to include a short discussion on what type of privacy the method achieves. We have also added this clarification to the introduction and now use the term 'data confidentiality' throughout the manuscript in order to refer to the notion of clients not sharing their data.
> >
> > The revised camera ready version has been uploaded. Thanks!
> >
> > Best,
> > The Authors

---

### Decision · Action_Editor_FN1t · 2023-10-20

**Recommendation:** Accept as is

**Comment:**

The paper proposes an interesting application of cryptographic primitives to a federated learning problem.
As noted by some reviewers, the techniques are somewhat straightforward, and might not be broadly applicable to other tasks.
The proposes approach is also limited to cryptographic data confidentiality, and doesn't provide stronger notions of privacy such as differential privacy.
Nevertheless the reviewers agreed this is an interesting direction worth accepting.

**Audience:**

Yes, there is a large audience interested in privacy and federated learning

**Claims And Evidence:**

The paper applies standard cryptographic techniques to the problem of label propagation.
Some reviewers were asking for additional theoretical guarantees, which the proposes scheme does inherit directly from these crypto primitives. The rebuttal clarified this.

One thing that could be good for the authors to discuss in a limitation section in the paper is the distinction between data confidentiality (which this paper achieves) and more general and powerful notions of privacy (akin to differential privacy) which it does not.